# Correction of Spacecraft Magnetic Field Noise: Initial Korean Pathfinder Lunar Orbiter MAGnetometer Observation in Solar Wind

**DOI:** 10.3390/s23239428

**Published:** 2023-11-26

**Authors:** Junhyun Lee, Ho Jin, Khan-Hyuk Kim, Hyeonhu Park, Wooin Jo, Yunho Jang, Hyeonji Kang, Eunhyeuk Kim, Young-Jun Choi

**Affiliations:** 1School of Space Research, Kyung Hee University, Yongin 17104, Republic of Korea; jhlee654321@khu.ac.kr (J.L.); khan@khu.ac.kr (K.-H.K.); hoo7781@khu.ac.kr (H.P.); whdndls99@khu.ac.kr (W.J.); toujour@khu.ac.kr (Y.J.); hjkang17@khu.ac.kr (H.K.); 2Korea Aerospace Research Institute, Daejeon 34133, Republic of Korea; eunhyeuk@kari.re.kr; 3Korea Astronomy and Space Science Institute, Daejeon 34055, Republic of Korea; yjchoi@kasi.re.kr

**Keywords:** Korean Pathfinder Lunar Orbiter mission, spacecraft-generated disturbance, gradiometer technique, initial KMAG observation

## Abstract

The Korean Pathfinder Lunar Orbiter (KPLO)-MAGnetometer (KMAG) consists of three triaxial fluxgate sensors (MAG1, MAG2, and MAG3) that measure the magnetic field around the Moon. The three sensors are mounted in the order MAG3, MAG2, and MAG1 inside a 1.2 m long boom, away from the satellite body. Before it arrived on the Moon, we compared the magnetic field measurements taken by DSCOVR and KPLO in solar wind to verify the measurement performance of the KMAG instrument. We found that there were artificial disturbances in the KMAG measurement data, such as step-like and spike-like disturbances, which were produced by the spacecraft body. To remove spacecraft-generated disturbances, we applied a multi-sensor method, employing the gradiometer technique and principal component analysis, using KMAG magnetic field data, and confirmed the successful elimination of spacecraft-generated disturbances. In the future, the proposed multi-sensor method is expected to clean the magnetic field data measured onboard the KPLO from the lunar orbit.

## 1. Introduction

The Korean Pathfinder Lunar Orbiter (KPLO) MAGnetometer (KMAG) instrument was designed to measure the intrinsic lunar magnetic field and the global response of the Moon to magnetic fields originating from the Sun and the Earth, using three triaxial magnetometers mounted on a 1.2 m long boom. In magnetic field observations during space exploration missions, it is well known that a satellite’s power, propulsion subsystem, and onboard instruments generate their own magnetic fields, producing magnetic interference.

The spacecraft’s magnetic interference field is a major noise source, which is displayed on a magnetometer instrument during payload observation. Thus, it is difficult to perform accurate in situ measurements of intrinsic magnetic fields in space [1,2,3]. To avoid the magnetic disturbance produced by spacecraft on previous lunar missions, the boom on the Lunar Prospector and Kaguya is 2.5 m [4] and 12 m [5,6] long, respectively. Another way to minimize spacecraft-generated magnetic fields is to clean up the spacecraft magnetically. This magnetic cleaning procedure is complex, and there are many constraints involved in carrying out the design and manufacture of the other onboard instruments [7]. Even if the magnetic cleaning process is applied during the development stage, it is not easy to remove this noise. Therefore, the magnetic field instrument usually uses a specific configuration wherein the magnetometer is installed on a long boom.

When a short magnetometer boom was used in previous missions, a multi-sensor method, employing the gradiometer technique and principal component analysis, was applied to the magnetic field analysis to separate the spacecraft-generated magnetic fields. The first study using the gradiometer technique involved fitting a dipole to a spacecraft-generated field, as proposed by Ness et al. [2]. This technique was successfully applied to magnetic field data from Mercury obtained by the Mariner 10 satellite, equipped with a 5.8 m long boom [8]. Neubauer et al. [3] suggested that using multiple (three or more) magnetometers is more accurate than using two magnetometers to correct a spacecraft-generated field. Pope et al. [9] successfully identified the magnetic interference generated from the Venus Express and eliminated the satellite interference in the time domain by applying the gradiometer technique. The gradiometer technique has been applied to remove magnetic interference during many space missions [10,11]. Recently, Constantinescu et al. [12] used the maximum variance analysis and gradiometer technique to reduce spacecraft-generated magnetic field disturbances from magnetic field measurements observed by the geosynchronous GK2A satellite [13]. In addition, to check the stability of the magnetically cleaned data from the GK2A satellite, the procedure of magnetic cleanliness by using the gradiometer technique is conducted once every week [12].

In the case of the KMAG instrument, the boom length is 1.2 m due to the KPLO design constraints; therefore, the KMAG has three magnetometers installed to find a noise source using multi-sensor techniques. The KMAG magnetic field measurements in solar wind were compared with the Deep Space Climate Observatory (DSCOVR) magnetometer observation dataset to verify the stability and reliability of the KMAG data. The KMAG data were found to be very stable and had good reliability. However, we confirmed that an unwanted spacecraft interference field existed in the observation data, as we expected. To identify a spacecraft-generated magnetic field, we used the dataset from the three magnetometers installed inside the boom and a magnetoresistance sensor inside the spacecraft body [14]. In general, a spacecraft-generated magnetic field is reduced according to the distance from the center of the spacecraft, and the dependence on distance is also confirmed under the condition of multiple magnetic sources inside the spacecraft, as suggested by Park et al. [15].

In this study, we clearly identified the spacecraft-generated magnetic field produced by the KPLO and applied a multi-sensor method, employing the gradiometer technique and principal component analysis, and using KMAG magnetic field data. Spacecraft-induced magnetic fields in solar wind were successfully eliminated by using a multi-sensor method.

The remainder of the paper is organized as follows: Section 2 describes the instrumental information about the KPLO satellite and the KMAG sensor. Section 3 outlines the proposed principal component gradiometer technique and the modification of the dataset. Section 4 describes how the proposed method eliminates spacecraft-generated disturbances in KMAG data, and Section 5 concludes the study.

## 2. Instrumentation: KMAG

The KPLO MAGnetometer (KMAG) is one of six scientific payload instruments on the Korean Pathfinder Lunar Orbiter satellite mission. The scientific objective is to investigate lunar magnetic anomalies and electromagnetic properties within the Earth’s magnetosphere and interplanetary magnetic field. The technical objective is to investigate the space operation of a multi-sensing technique and fluxgate sensor. The KMAG system has two sets of magnetometers, MAG and anisotropic magnetoresistive (AMR) sensors, within the fluxgate magnetometer control electronics (FCE) unit.

The MAG consists of three triaxial fluxgate magnetometers inside a 1.2 m boom, made using a carbon fiber-reinforced plastic tube. Figure 1 illustrates the placement of the three magnetometers: MAG1 is positioned at the tip of the KMAG boom, and MAG2 and MAG3 are installed closer to the spacecraft body. Their distance from the spacecraft body is 1.20, 0.95, and 0.58 m, respectively. Each sensor has a measurement range of ±1000 nT, with less than 0.2 nT resolution at a sampling rate of 10 Hz. The magnetic noise level remains lower than 50 pT/Hz^0.5^ at 1 Hz. During the ground calibration process, the orthogonality and linearity of each magnetometer are confirmed, ensuring that errors are below 0.9° and 0.05, respectively. The KMAG boom is deployed at an angle of 135° from the top floor of the spacecraft. The sensor coordinate system for each magnetometer is structured so that the *Z*-axis aligns with the direction from the boom tip to the spacecraft body. The *X*-axis is placed counterclockwise, perpendicular to the *Z*-axis, and the *Y*-axis completes a right-handed orthogonal set.

The FCE unit controls the overall KMAG system, including commands, data handling, and communication with the spacecraft. This unit consists primarily of the analog board (AB), digital board (DB), onboard computer (OBC), and low-voltage power supply (LVPS). Notably, the OBC contains a low-resolution AMR sensor, namely a Honeywell HMC1053 sensor, which measures unusually large magnetic field signals from the spacecraft body. The original purpose of the AMR sensor was ground test assistance. However, the AMR sensor now contributes to eliminating the magnetic interference associated with spacecraft operations. The AMR sensor operates within a measured range of ±60 μT, with a resolution of 100 nT at a sampling rate of 10 Hz. However, as the AMR sensor is susceptible to temperature variations, it is necessary to monitor the temperature in order to calibrate the magnetic field data accurately. For this study, the data from the AMR sensor was transformed to match the coordinate system of the MAG sensors. Table 1 summarizes the specifications for each magnetometer integrated into the KMAG.

## 3. Clearance Technique for Spacecraft-Generated Magnetic Disturbances

During space missions, the operation of electronic devices inside the spacecraft can generate magnetic fields that interfere with the measurement of the ambient magnetic field. Although ground calibration can be conducted to remove the magnetic interference by the spacecraft, unexpected disturbances can remain for the operating period of the spacecraft after launch. These spacecraft-generated disturbances pose a significant challenge for accurate magnetic field measurements and necessitate the implementation of proper mitigation strategies. Therefore, to obtain reliable magnetic field measurements, it is imperative to monitor the spacecraft’s operational phase and implement in-orbit clearance procedures to address the disturbance signals.

### 3.1. Elimination of Artificial Disturbance

In general, the magnetic field measured by the magnetometer is described by the integration of the ambient magnetic field *B*(*t*), the spacecraft-generated magnetic field *b^i^*(*t*), and the constant specific disturbance on the micro-time scale, called the sensor offset, *Z^i^*(*t*). The summation of the magnetic fields measured by sensor *i* is expressed as:(1)$$B^{i}=B\left(t\right)+\overset{N}{\underset{q=1}{\sum }}b_{q}^{i}\left(t\right)+Z^{i}\left(t\right)$$
where the index q indicates the number of sources for the spacecraft-generated disturbance to *N*. The sensor offset is a constant value for each sensor in Equation (1). The first and second terms on the right-hand side can be considered as the dipole field. The dipole field characterized by magnetic moment *M* is:(2)$$B_{dipole}=\frac{\mu_{0}}{4\pi }\left[\frac{3\vec{r}\left(\vec{r}\cdot M\right)}{{\left|r\right|}^{5}}-\frac{M}{{\left|r\right|}^{3}}\right]$$
where $\mu_{0}$ is the magnetic permeability of free space and $\vec{r}$ is the position vector between the measuring point and the magnetic moment. Considering the negligible difference in the position vectors between the magnetic moment of the ambient field *B*(*t*) for each magnetometer, it is assumed that the simultaneously observed ambient field is identical for all sensors. However, in the presence of interference, denoted as *b*(*t*), the position vectors for each sensor exhibit significant differences compared to the ambient field scenario. As a result, subtracting the measurements from two sensors positioned at distinct locations eliminates the ambient field term *B*(*t*) from Equation (1), while retaining the terms associated with interference *b*(*t*) and the sensor offset *Z*(*t*):(3)$$\begin{array}{l} \Delta B^{ij}\left(t\right)=B^{i}\left(t\right)-B^{j}\left(t\right)\\ =\overset{N}{\underset{q=1}{\sum }}b_{q}^{ij}\left(t\right)+\Delta Z^{ij}\left(t\right) \end{array}$$

While the artificial disturbance generated by spacecraft occurs in a short time range of minutes, the offset from all the sensors fluctuates over a long period of hours to days. The variance-based variation in the magnetic field data from each sensor is utilized to extract the artificial disturbance and correct it. Thus, the offset represented by *Z*(*t*) in all equations can be treated as a constant and can be ignored in this correction process. Consequently, the difference in measurements between the two sensors can be simplified as solely the disparity in the interference measured by the sensors. Although the position of the artificially disturbed source should be defined to eliminate interference, it is impossible to track all positions of the source accurately. However, assuming that each disturbance magnetic field is the dipole field from Equation (2), the summation of all the disturbances shows a dependence on the distance of the sensors from the spacecraft body, similar to the centered dipole approximation. In previous studies, by using multi-sensors, it is reported that the artificial disturbance can be defined with dependence on the sensor position, without considering the number and position of the disturbance sources [2,15]. Therefore, if the interference measured by each sensor shows a dependence on the distance of the sensor from the spacecraft when the spacecraft is maneuvering, the artificial disturbance can be efficiently removed by using the variances between the two sensors. From these assumptions, multiple disturbed sources can be considered as a single disturbed source. Equations (1) and (3) are used to reduce the sources to the form of a single disturbed source:(4)$$B^{i}=B\left(t\right)+b^{i}\left(t\right)+Z^{i}\left(t\right)$$
(5)$$\Delta B^{ij}\left(t\right)=\Delta b^{ij}\left(t\right)-\Delta Z^{ij}\left(t\right)$$

In addition, when the spacecraft temporarily maneuvers, the transient variation of the variance in the measurement, which is affected by the artificial disturbance, is much greater than the variation in the ambient field at that time. This variance could be used to determine the degree of significant disturbance for each direction. Using variance analysis, it is possible to define the maximum variance direction corresponding to the artificial disturbance at the sensor position and isolate it [16,17]. The maximum variance direction of the measurement represents a more robust component of the disturbance than other regular and random disturbances. We applied principal component analysis (PCA) to define the direction of maximum variance, called the variance principal system (VPS).

The interference removal process can be outlined as follows: (1) restrict the interference period associated with spacecraft maneuvering; (2) determine the rotation matrix of *B^i^*(*t*) and Δ*B^ij^*(*t*) to transform from the sensor coordinates system described in Figure 1 to the VPS system, extracting the component with the maximum variance; (3) correct the interference of *B^i^*(*t*) only on the maximum variance direction by utilizing the variance. A schematic flowchart of the correction process to remove the spacecraft-generated disturbance is shown in Figure 2.

With the VPS, by using the measurement from sensor *j*, the relation-corrected sensor *i* measurement can be written as:(6)$$B_{L}^{1,i}=R_{1k}^{i}B^{0,i}-\alpha^{0,ij}R_{1k}^{ij}\Delta B^{0,ij}$$
(7)$$B_{M}^{1,i}=R_{2k}^{i}B^{0,i}$$
(8)$$B_{N}^{1,i}=R_{3k}^{i}B^{0,i}$$
where *R* is the rotation matrix from the coordinates in the sensor system toward the variance principal system, represented by the L–M–N axes to distinguish them from the x–y–z axes in the sensor coordinates in this section. In this coordinate system, the L, M, and N axes are parallel to the maximum, intermediate, and minimum variance directions, respectively. The subscript *nk* of R indicates that row *k* of R is the projection of *n* along the axes defined by the L, M, and N directions. The superscript 0 in Equations (6)–(8) indicates the uncorrected measurements in the VPS system, and the superscript 1 means the first-order correction. Moreover, $\alpha^{0,ij}$ is the scaling factor. Assuming the disturbance difference between *B^i^*(*t*) and Δ*B^ij^*(*t*) is due to the sensor position, the relationship between *B^i^*(*t*) and Δ*B^ij^*(*t*) is proportional and is reflected in the scaling factor. Therefore, to eliminate the disturbance signal in *B^i^*(*t*) by using their proportionality, the $\alpha^{0,ij}$ factor is considered as the ratio of the variance in the measurements:(9)$$\alpha^{0,ij}=\pm \sqrt{\frac{Var\left({\left(B^{0,i}\right)}_{L}\right)}{Var\left({\left(\Delta B^{0,ij}\right)}_{L}\right)}}$$

Since the orientation for the maximum variance is uncertain in Equation (9), the ± sign indicates that the positive direction on the L-axis is arbitrary. After performing this process, the orientation is determined by selecting the minimized correlation coefficient between the corrected measurement and the difference. Hence, Equations (6)–(8) can be simplified for the sensor’s coordinate system with the correction matrix *A*, including the scaling factor and rotation matrix:(10)$$A^{0,ij}=-\alpha^{0,ij}{\left({\left(R^{i}\right)}^{-1}\right)}_{kx}{\left(R^{0,ij}\right)}_{xl}$$
(11)$$B^{1,i}=B^{0,i}+A^{0,ij}\Delta B^{0,ij}$$

Briefly, with regard to this method, only the measurement corresponding to the maximum variance direction is corrected by using the scaling factor, based on the assumption that the artificial disturbance transiently causes the largest variance.

### 3.2. Mitigation of Amplified High-Frequency Noise

In the magnetic disturbance correction process, there is an instance where the difference between *B^i^* and *B^j^* becomes small. For example, in cases where MAG1 needs to be calibrated with MAG3 data instead of MR data, the difference Δ*B^0,ij^* between the MAG1 and MAG3 data in Equations (9) and (11) is relatively more minor than when using MR data. Since Δ*B^0,ij^* depends on the distance between both sensors, the variance in (Δ*B^0,ij^*)_L_ between two sensors located close together decreases. The reduced variance leads to an increase in the absolute value of $\alpha^{0,ij}$ in Equation (9). The increased $\alpha^{0,ij}$ factor can amplify the data in times when disturbance does not occur, and the amplified data comes out as high-frequency noise on the axes (B_L_) in the maximum variance direction. To prevent interference from the amplified high-frequency noise affecting the outcome, the mitigation process uses uncorrected and corrected values over the direction of maximum variance only before applying the rotation matrix *r^i^*. Our strategy is to select a time interval that is not affected by disturbances generated by the spacecraft for mitigated correction. Then, we calculate and average the variance for an arbitrary time window over both values. By using the averaged variance values, we compute the ratio as follows:(12)$$r^{i}=\sqrt{\frac{Var\left(R_{1k}^{i}B^{0,i}\right)}{Var\left({\left(B^{1,i}\right)}_{L}\right)}}\;\;,\;\;\;\left({\left(B^{0,i}\right)}_{L}=R_{1k}^{i}B^{0,i}\right)$$
(13)$$B_{L,\;\;after}^{1,i}=r^{i}\times B_{L,before}^{1,i}$$

Finally, the high-frequency noise is mitigated by multiplying the ratio by the corrected value in the maximum variance direction only, as in Equation (13).

## 4. Results

Before the lunar orbit insertion, the KMAG team conducted a calibration procedure to account for the disturbances generated by the spacecraft maneuvering and calculated the daily zero-offset value for the dataset. Two days after deploying the KMAG boom, an unexpected disturbance appeared in the KMAG observation data from the spacecraft maneuvering. Since the zero-offset calculation uses the method that minimizes the variance in the field magnitude, the measured magnetic field containing disturbances made the result of the zero-offset calculation inaccurate. This section outlines a case concerning the clearance results and process associated with a spacecraft-generated disturbance on 6 August 2022 at a position just outside the Earth’s bow shock.

### 4.1. Comparison of DSCOVR FGM and KMAG

Figure 3 displays the time series of the magnetic field measured by the KPLO MAG1 and DSCOVR FGM using the GSE coordinate system and the operational information from the KPLO satellite related to the attitude and interior electronics on 6 August 2022. All the components of the magnetic field data from MAG1 are displayed in Figure 3a–c; 10 nT is added after subtracting the daily averaged value to compare the data with the DSCOVR FGM measurements. It is worth noting that the daily averaged value from MAG1 does not represent its daily zero-offset value. Additionally, we confirmed that there were no dynamic solar wind events during the same period. As shown in Figure 4, the DSCOVR satellite is positioned at the L_1_ point (223.9 R_E_, −14.9 R_E_, −24.4 R_E_) in the geocentric solar ecliptic (GSE) coordinate system, while the KPLO is located just outside the Earth’s bow shock (32.9 R_E_, 8.7 R_E_, −8.2 R_E_), with R_E_ referring to the Earth’s radius. According to Richardson and Paularena [18], if the difference in distance of both satellites in the GSE Y–Z plane is less than 50 R_E_ in the interplanetary medium, it is possible to measure the same solar magnetic field. Therefore, since the distance between the DSCOVER and KPLO satellites is approximately 16 R_E_ on the Y–Z plane, the KMAG can observe a similar magnetic field to that measured by DSCOVR FGM. Nevertheless, the difference of ~191 R_E_ between the two satellites along the X_GSE_ direction causes a 40 min delay in solar wind propagation. The calculations for this time delay are applied to the KMAG measurements in Figure 3.

To evaluate the consistency of the magnetic measurements from each position within the interplanetary region, we conducted a comparison of the hourly averaged magnetic fields, as illustrated in Figure 5. Given the absence of significant interplanetary magnetic field (IMF) variations within a 1 h window during quiet solar wind conditions, we calculated the standard deviation based on 1 h boxcar averaged data without temporal overlaps. The relationship between the measurements shows notably high correlation coefficients of 0.77, 0.86, and 0.98 for the B_X_, B_Y_, and B_Z_ components, respectively, except for a 2 h interval starting from 04:00 UT. In addition, the background magnetic field, as measured by DSCOVR FGM, exhibits a standard deviation of less than 1 nT throughout the entire observation period. The MAG1 measurements show a comparable standard deviation of less than 1 nT over the same duration. However, there are instances of abrupt and substantial variations in the averaged magnitude and standard deviation, particularly for the B_X_ and B_Y_ components in the MAG1 measurements. These variations increase up to four times compared to the DSCOVR FGM measurements during the time intervals of 04:44 to 05:44 UT and 23:11 to 24:00 UT, which can be characterized as unknown disturbed periods.

This result suggests two perspectives. Firstly, the consistency of the observations between DSCOVR FGM and KMAG shows that KMAG is a reliable tool for magnetic field measurements. Secondly, considering that DSCOVR FGM can accurately observe the background magnetic field, the difference in the measurements between the two satellites for the unknown disturbed period implies that the KMAG measurements may have originated from other magnetic sources.

Thus, for the results in Figure 3a–c, it is considered that KMAG measured the magnetic fields related to different magnetic sources in the background field during the unknown disturbed period. To find the disturbed magnetic source, we conducted a survey of the satellite’s attitude data. Figure 3d confirms that the satellite’s Euler angle for all directions suddenly varied for about 12 min from 05:24 to 05:36 UT. Variations in the spacecraft’s Euler angles indicate changes in its attitude, such as spinning or flipping, which is strongly linked to the disturbed magnetic field. When the spacecraft’s attitude changes, the operation of the electronic equipment within its body is typically expected. However, this finding is inadequate to explain the measurement taken before 05:24 UT. To find the disturbance source, an additional investigation was conducted to determine the relationship between the spacecraft-generated disturbance and spacecraft maneuvering using the monitoring data from the valve drive electronics (VDE), as depicted in Figure 3e.

The identification of the VDE activation indicates that the circuit inside the spacecraft is carrying an electrical current relevant to the operation of the thrusters, as indicated by as turned-on or -off signals depending on whether the electronic devices are activated or not. The VDE switching signal clearly coincides with the time of the abrupt variations in the KMAG observation data, or the unknown disturbance period. For example, the occurrence of the VDE switching-on signal for 1 h from 04:44 UT accurately coincides with the abrupt variations in the KMAG measurements. Consequently, these abrupt variations in the measurements, as depicted in Figure 3a–c, indicate that the measurement includes artificial disturbances generated by the spacecraft.

### 4.2. Elimination of Artificial Disturbance

Figure 6 shows the measurements from the MAG1, MAG2, MAG3, and MR sensors on 6 August 2022, in the sensor reference coordinate system. Note that since the zero-offset calculation was not yet applied to each sensor’s data, the magnetic field observation in Figure 6 does not represent a realistic value for the background field. The MAG and MR measurements are expressed on the left and right axes, respectively, in units of nT. In Figure 5a, the abrupt changes in the B_X_ component from all MAG sensors, as visible in Figure 3, are not shown from 04:44 to 05:44 UT. However, there is a decrease of approximately 40 nT in the measured B_X_ for the MR sensor. As shown in Figure 6b, although the B_Y_ from the MR sensor gradually decreases over time by about 100 nT, the MAG and MR sensor measurements show no evidence of the artificial disturbance toward the Y direction. The B_Z_ components from the MAG and MR sensors in Figure 6c simultaneously show a step-like shaped disturbance from 04:44 to 05:44 UT.

The magnitude of this disturbance increased suddenly compared to the period before the disturbance, by ~4.5 nT for MAG1, ~6.1 nT for MAG2, ~7.0 nT for MAG3, and ~60.0 nT for MR. These varying disturbance magnitudes are dependent on the distance of each sensor from the center of the spacecraft. The relationship between the disturbance magnitude and the sensor position obviously indicates the presence of an additional magnetic source within the spacecraft body. By analyzing the readings from the MR sensor, which detected a disturbance in the B_X_ and B_Z_ components, it can be deduced that the magnetic disturbance was caused by a source with a magnetic moment aligned parallel to the B_Y_ component. Hence, the detection of magnetic disturbances with all sensors exhibiting distance-dependent variations in the same directional components again strongly supports that these disturbances originated from within the satellite’s interior. Figure 6d shows a zoomed-in view of the 230 min interval in regard to the BZ component of the magnetic field measurement for all the KMAG sensors, indicated by the black bar in Figure 6c. It is worth noting that the measurements from all the MAG sensors were adjusted by subtracting the daily average, except for the MR measurement. Spike-like disturbances are observed just before and after the step-like disturbance in the measurements obtained from all the MAG sensors. The spike-like disturbances have a shorter interval of ~2 min compared to the step-like disturbances. However, since the MR sensor measurement was averaged in the preprocessing, the spike-like disturbance is not presented in the MR sensor measurement. For this reason, the MR sensor measurement is useful for the removal of the step-like disturbance only.

Therefore, our strategy is to preferentially eliminate the step-like disturbance from the data from each MAG by using the MR measurements, and then remove the spike disturbance using only the MAG data. The time interval for the disturbance elimination is selected from 04:30 UT to 06:00 UT to isolate the targeted step-like disturbance. Using the gradiometer technique mentioned in Section 3.1, we determined the direction for the maximum variance to find the scaling factor and the rotation matrix. To compare the most evident difference from the magnetic source inside the spacecraft body, we used the data from the inboard sensor and the outmost sensor on the spacecraft, MR and MAG1.

As shown in Figure 7, the step-like disturbance was extracted using the difference Δ*B*^0,*ij*^ between the MAG1 and MR sensors for each component in the VPS coordinate system, using the PCA method. Specifically, Figure 6a–c shows these components, which were rotated to align with the directions of maximum, intermediate, and minimum variance. These components correspond to the LMN coordinate components explained in Section 3.1. In Figure 7a, the step-like disturbance is also extracted for the selected time interval and the second disturbance after 23:00 UT. The extraction of both step-like disturbances related to the maximum variance components at the same time implies that both disturbances can be eliminated in the first-order correction process, simultaneously. Figure 7b shows the largest change of ΔB_Y_, by ~140 nT across all time periods, while Figure 6a shows that ΔB_X_ changes by ~90 nT over the entire period of the data. However, it is important to note that the variance direction is determined based on the difference values of ΔB_X_ ~65 nT, ΔB_Y_ ~40 nT, and ΔB_Z_ ~20 nT for the isolated time interval. In other words, the rotation matrix used to transform to the VPS coordinates only reflects these difference values in the isolated time range. Figure 7c shows the minimum variation in the same interval, representing the minimum variance direction. Since the MR measurement initially includes high-frequency noise, the result obtained by subtracting both sensor measurements reflects this noise for all the components, as demonstrated in Figure 7a–c. However, in the first-order correction process, the high-frequency noise can be ignored due to the significantly more significant variance in the step-like disturbance.

The initial measurement and the first-order corrected results are shown using the initial VPS coordinate system in Figure 8, as black and red lines, respectively, with the mean value subtracted. In Figure 8a, the targeted step-like disturbance occurring between 04:45 and 05:44 UT is successfully eliminated in the maximum variance direction, while the spike-like disturbance still persists. In addition, despite the correction process, the corrected result after 05:44 UT differs from the uncorrected measurement. According to Constantinescu et al. [12], this discrepancy arises because spacecraft-generated disturbances continue to affect the magnetic field measurements, even after the maneuver has concluded. In accordance with Equations (6)–(8), it is important to note that the intermediate and minimum variance direction components in Figure 8b,c do not influence this correction step.

Figure 9 shows a spike-like disturbance still present in the first-order corrected results of the measurements from the three MAG sensors. We selected an isolated time range of 6 min from 04:42 to 04:48 UT to decouple the spike-like disturbances after the first-order correction. The disturbance in this period, shown in Figure 9, corresponds to the spike just before the step-like disturbance. The spike-like disturbance in relation to the Bx components using the VPS coordinates, which is the same as the B_L_ component in Section 3, is extracted from the first-order corrected results of the three MAG sensors using the calculated rotation matrix, as depicted in Figure 9a. The minimum peak magnitude of each disturbance on the Bx component is −7.5 nT for MAG1, −9.8 nT for MAG2, and −13.8 nT for MAG3. Similar to the case of the step-like disturbance, this result also indicates that the magnitude of these disturbances is well suited to the differences in the distance between the magnetometers. On the other hand, the other components in Figure 9b,c show only high-frequency noise initially included in the measurement. The high-frequency noise tends to have a larger amplitude in the MAG3 measurement than for MAG1 and MAG2. This tendency for noise is also shown in Figure 9a. Therefore, we can expect that the different amplitudes of high-frequency noise affect the results after the second-order correction, according to Equations (6)–(8), when we use only the MAG sensors to conduct the second-order correction.

In the second-order correction, the initially corrected results are adjusted by the maximum variance direction to extract the spike-like disturbance. Since the MR sensor cannot measure spike-like disturbances, the measurement from the MAG3 sensor is used to eliminate the disturbance. The MAG3 sensor is located farthest from the MAG1 sensor. The second-order correction result for the MAG1 measurement over the remaining spike-like disturbance is plotted in Figure 10. In the VPS coordinate system for the first-order correction result, the first-order and second-order correction results are depicted as black and red lines, respectively, using the same format as Figure 8. As in the case of Figure 8, the second-order correction is conducted only for the component in the maximum variance direction. In addition, although there are two spike-like disturbances just before and after the step-like disturbance, we perform the correction for the preferentially observed spike-like disturbance with time as a target. In Figure 10a, the second-order correction process eliminates the targeted spike-like disturbance from 04:44 to 04:47 UT, along the maximum variance direction. However, the initial result, as indicated by the magenta line, exhibits a broader range of variation compared to the first-order corrected result. The initial outcome for the second-order correction displays an amplitude change of up to ~2 nT as high-frequency noise, surpassing the variation observed in the first-order corrected result. This high-frequency noise can be considered an artificial outcome that occurs during the correction process, and it is mitigated according to the method outlined in Section 3.2. Based on the calculations using Equation (12), the ratio of the standard deviation for the first-order and initial second-order correction measurements was determined. The calculation was carried out for the period without spacecraft-generated disturbance, corresponding to the time interval from 00:00 UT to 04:00 UT and from 08:00 UT to 21:00 UT. The calculated ratio is approximately 0.32. The red line illustrates the final result, which mitigates high-frequency noise by multiplying the initial correction result by the calculated ratio.

Figure 11 shows the DSCOVR FGM measurement and the corrected MAG1 measurement using the GSE coordinate system. Both measurements are subtracted from their daily average. To compare the two measurements, the DSCOVR magnetic field data was time shifted by 40 min over the KMAG observation. The time shift was determined by identifying the peaks in the magnetic field data collected by the two spacecraft over 4 h starting at 10:00 UT. In addition, +7 nT and −7 nT are, respectively, added to the uncorrected KMAG data and the DSCOVR FGM data in each panel, to compare with the corrected magnetic field data. On the B_X_ and B_Y_ components in Figure 11a,b, indicated by a black bar, the step-like disturbance from 04:44 to 05:44 UT for the raw measurements (gray) is successfully eliminated in the corrected data (red). Although the scaling matrix *A*^0,*ij*^ in Equation (10) is applied over all the time ranges, only the disturbance is clearly removed. In addition, another step-like disturbance exists from 23:20 UT, indicated by a gray bar. This disturbance is removed after the correction process. Considering the switching-on signal for the VDE and the magnetic field components B_Y_ and B_Z_ related to the disturbance in Figure 2, it can be seen that the two disturbances under the black and gray bars share the same generating source. For this reason, even if the scaling matrix is only calculated for the time range indicated by the black bar, the matrix can eliminate the disturbance for the time range of the gray bar at the same time. As a result, Figure 11 shows good agreement between the DSCOVR and KMAG observations.

## 5. Summary and Conclusions

We described the data correction method for the KMAG magnetometer to remove the magnetic disturbances coming from the spacecraft. It is crucial to remove any disturbances to accurately determine changes in the background magnetic field, which is necessary to calculate the zero offset or identify magnetic events on the Moon. Therefore, the disturbances have to be eliminated during the initial data processing phase before the zero-offset calculation. Based on our observations, we confirmed that the disturbance generated by the spacecraft is related to its maneuvering, as indicated by the Euler angle and VDE data. When the KMAG’s three sensors, MAG1, MAG2, and MAG3, are sensing spacecraft-generated disturbances, the magnitude of the changes in the disturbances depends on how far away the sensor is from the spacecraft body. The spacecraft-generated disturbance that occurred on 6 August 2022 consisted of two types of disturbances, step-like and spike-like. The final correction result shows that the use of the maximum variance gradiometer technique is reasonable to eliminate disturbances originating from spacecraft operations. By assuming that the disturbance has a greater variance than the variation in the background field, principal component analysis can be used to arrange the magnetic disturbance in the direction with the maximum variance. We set the isolated time range for each type of disturbance and addressed them sequentially. The time range for applying the technique to address spacecraft disturbance can vary depending on the duration of the maneuvering period. This applies not only to the current event, but also to future events. In the correction process, filtered data from the MR sensor is used to remove the step-like disturbance that shows the most significant difference. Since the modified MR data no longer exhibit a spike-like shaped disturbance, using the first-order corrected data of MAG1 and MAG3, which are located farthest from each other, eliminates the spike-like disturbance within the 2 min time range.

Consequently, it is a useful method for eliminating noise, as we confirmed that the corrected results for the KMAG data are in alignment with the DSCOVR magnetic field data. However, during the satellite’s mission, not only the satellite’s maneuvers but also the operation of the payloads on the satellite is expected. It is nearly impossible to track all sources of disturbance. It is important to note that the proposed correction procedure cannot be a one-time task. The elimination of magnetic disturbances from the spacecraft needs to be conducted every time the spacecraft or its parts are activated. For this reason, in future operations, the KMAG team plans to monitor the effect of satellites on magnetic field measurements during the lunar orbit phase. In addition, to address any disturbances caused by the spacecraft, the KMAG team will utilize the gradiometer technique and principal component analysis, continuously.

## Figures and Tables

**Figure 1 sensors-23-09428-f001:**
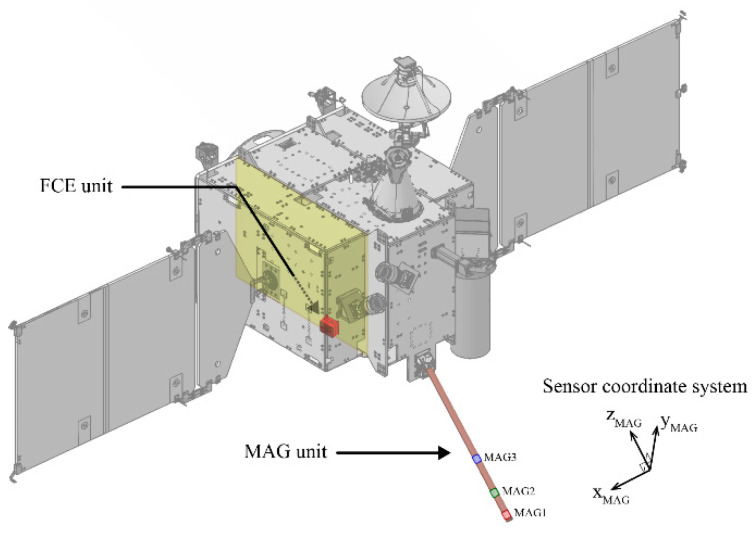
Overview of KMAG instrument on KPLO satellite, consisting of an FCE unit (red box), located inside the spacecraft body, and an MAG boom (orange), deployed externally. Within the boom are three magnetometers: MAG1 (red), MAG2 (green), and MAG3 (blue).

**Figure 2 sensors-23-09428-f002:**
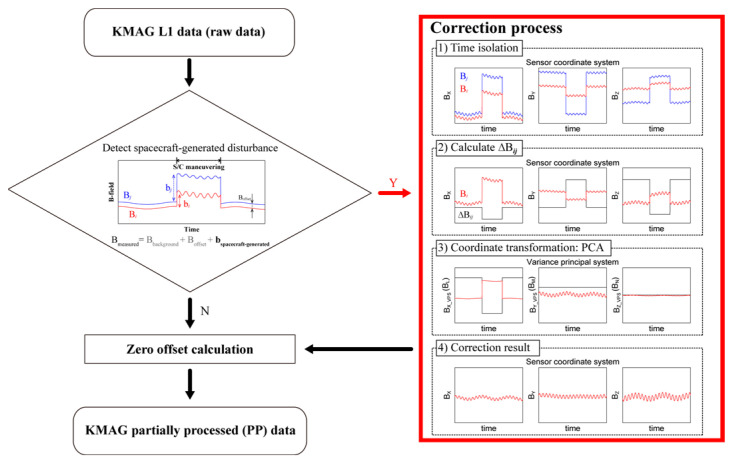
Flowchart describing the schematic approach used for the elimination of spacecraft-generated disturbance.

**Figure 3 sensors-23-09428-f003:**
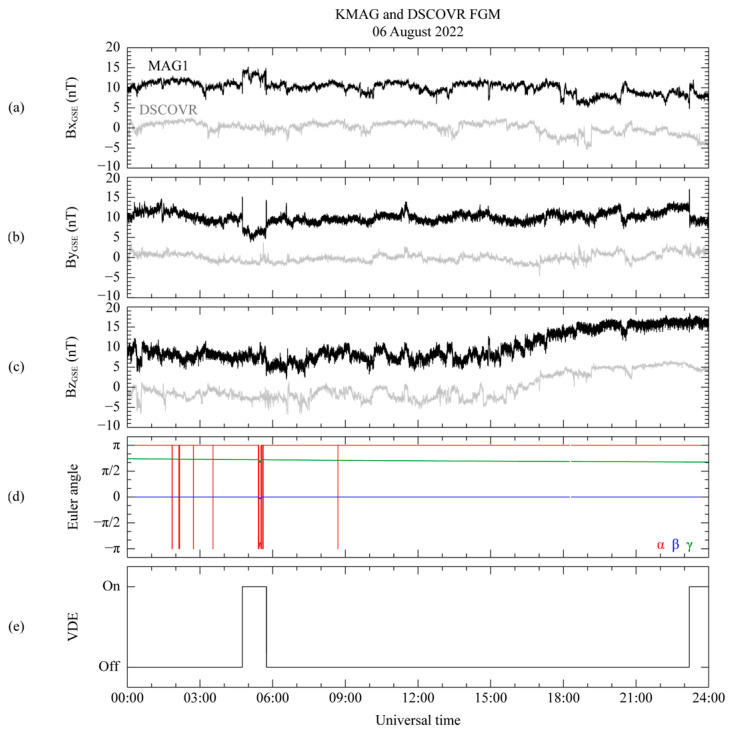
KMAG and DSCOVR FGM observations in geocentric solar ecliptic coordinate system on 6 August 2022 with KPLO satellite maneuvering information. (**a**–**c**) Magnetic field components B_X_, B_Y_, and B_Z_, respectively, from KMAG (black line) and DSCOVR (gray line). Each component is subtracted by its mean value. DSCOVR measurements are time shifted by 40 min. (**d**,**e**) Euler angle for each component (α: red; β: blue; γ: green) and valve drive electronics information for KPLO satellite, respectively.

**Figure 4 sensors-23-09428-f004:**
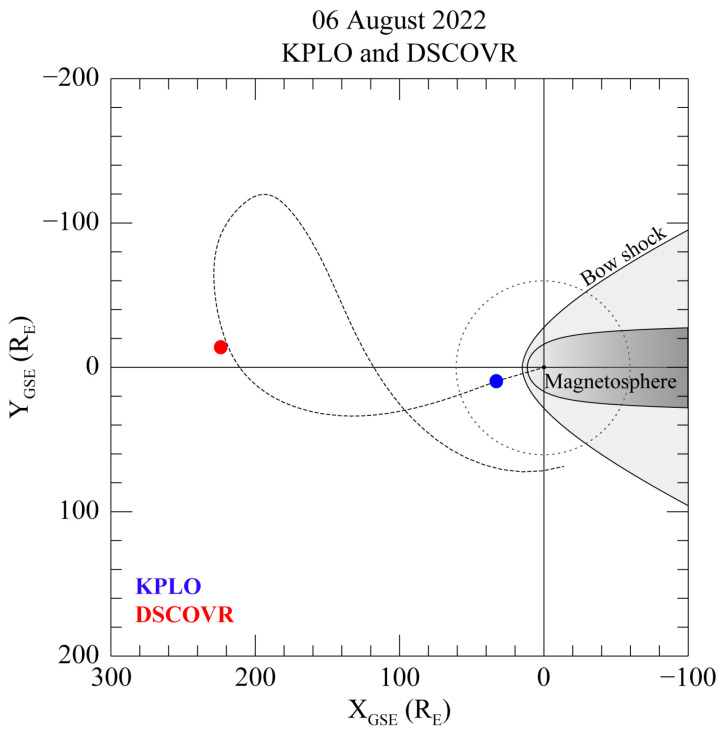
Positions of KPLO (blue) and DSCOVR (red) spacecraft, projected onto ecliptic plane using GSE coordinates. The lunar orbit is shown as a dotted circle, and the predicted orbit of the KPLO is represented by a dotted curved line.

**Figure 5 sensors-23-09428-f005:**
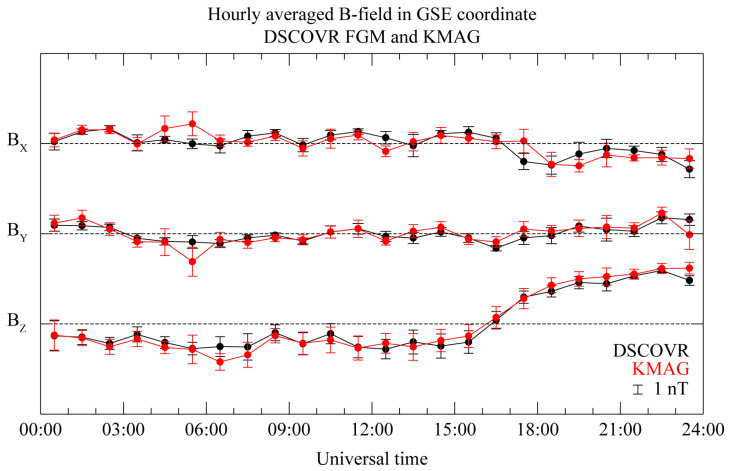
Temporal variation in hourly averaged magnetic field, subtracting daily mean value to support Figure 2. Black and red lines correspond to DSCOVR FGM and MAG1 sensor measurements, respectively; horizontal line represents 0 nT for each component.

**Figure 6 sensors-23-09428-f006:**
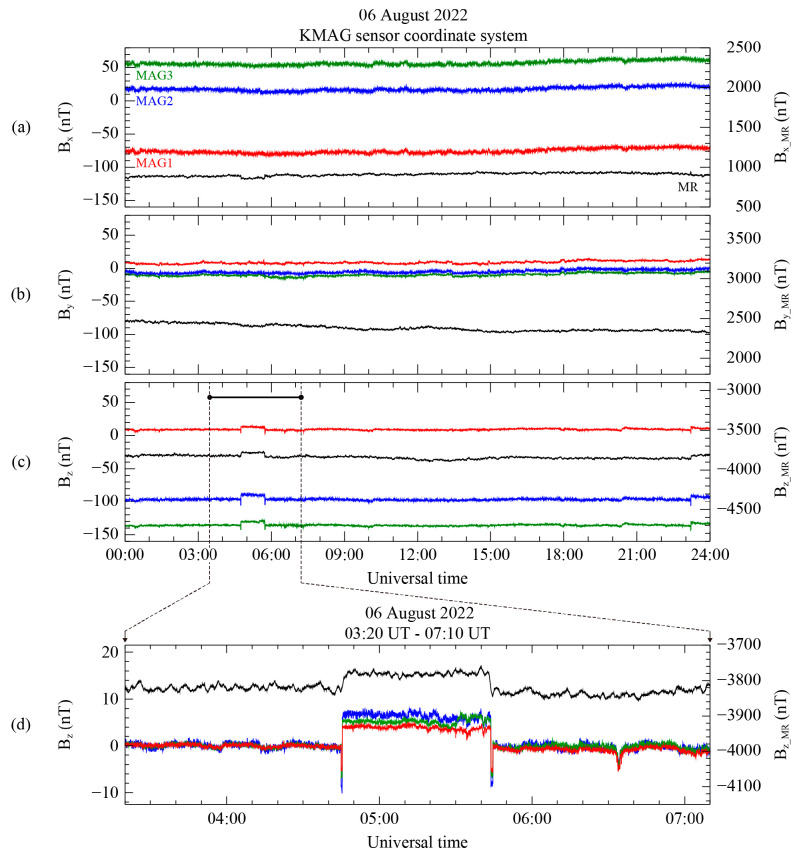
KMAG observations on 6 August 2022 using sensor frame coordinates. (**a**–**c**) B_X_, B_Y_, and B_Z_ components of the magnetic field, respectively, observed from MAG1 (red), MAG2 (green), MAG3 (blue), and MR (black). (**d**) Expanded view, in the time range from 03:20 to 07:10 UT, displays the B_Z_ component for all the sensors.

**Figure 7 sensors-23-09428-f007:**
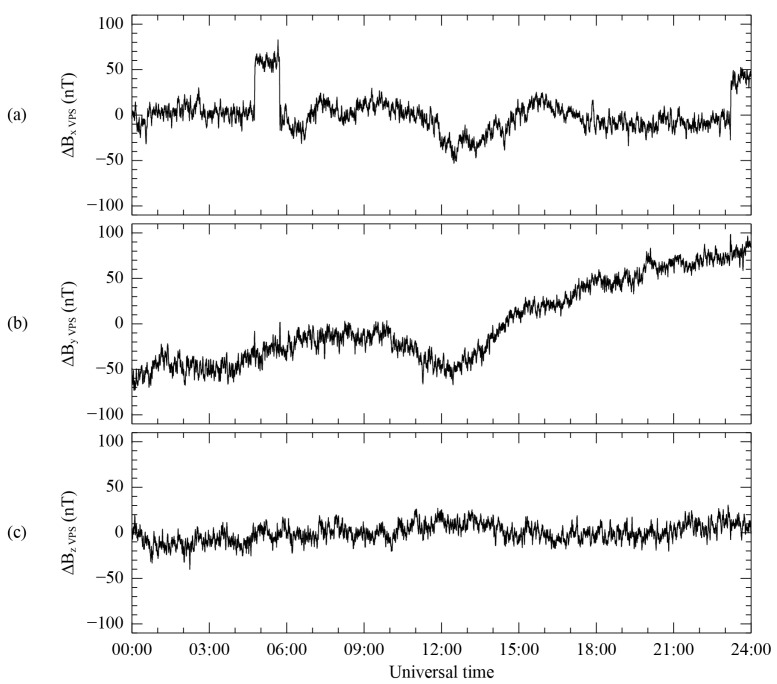
Difference ΔB^0,ij^ between MR and MAG1 before correction using the corresponding VPS coordinate system of ΔB0ij. Components B_X_ (**a**), B_Y_ (**b**), and B_Z_ (**c**) are aligned with the direction of the maximum, intermediate, and minimum variance, respectively. Mean values are subtracted from all the components.

**Figure 8 sensors-23-09428-f008:**
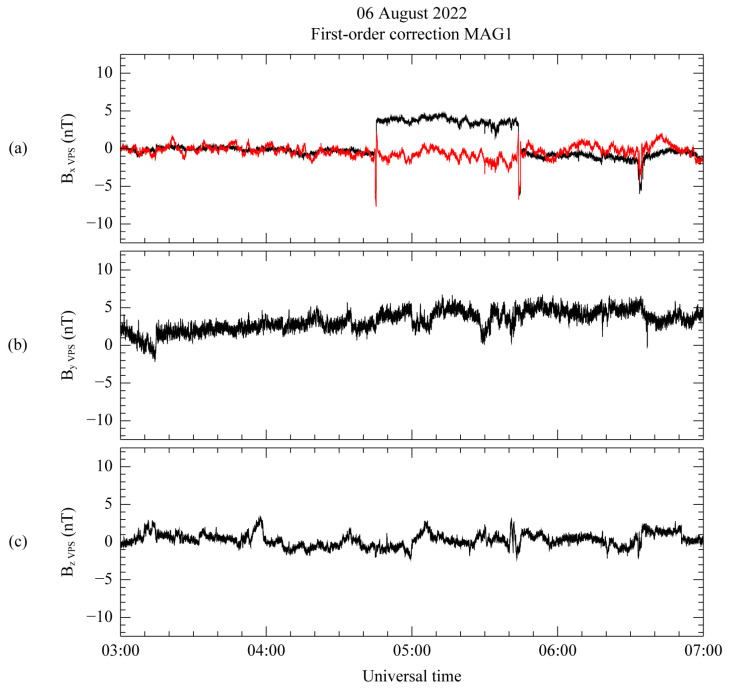
Uncorrected (black) and first-order corrected (red) measurements from MAG1 using VPS coordinates. Components B_X_ (**a**), B_Y_ (**b**), and B_Z_ (**c**) are oriented along the directions of maximum, intermediate, and minimum variance, in that order. The corresponding daily average for each component is subtracted from all the components.

**Figure 9 sensors-23-09428-f009:**
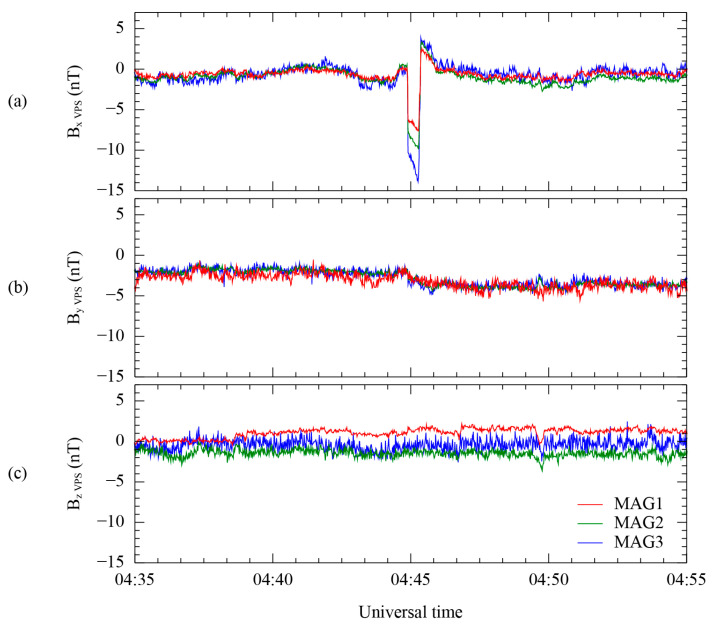
First-order corrected results for MAG1 (red), MAG2 (green), and MAG3 (blue) by using the MR for each VPS coordinate system. Components B_X_ (**a**), B_Y_ (**b**), and B_Z_ (**c**) are oriented in the direction of maximum, intermediate, and minimum variance. The corresponding daily average for each component is subtracted from all the components.

**Figure 10 sensors-23-09428-f010:**
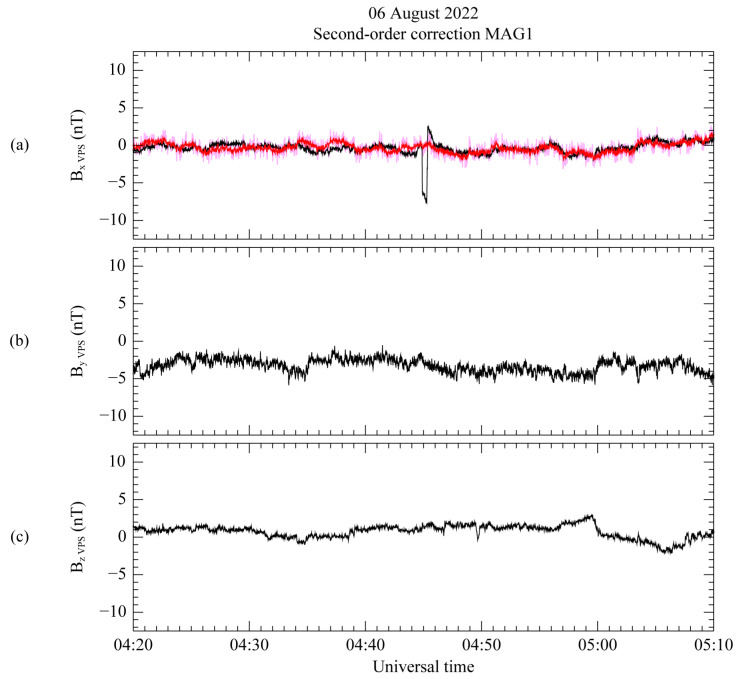
First-order corrected (black) and second-order corrected (red) measurements from MAG1 using VPS coordinates. Second-order corrected measurements without mitigating high-frequency noise are also displayed on the top panel as a magenta line. Components B_X_ (**a**), B_Y_ (**b**), and B_Z_ (**c**) are aligned with the directions of maximum, intermediate, and minimum variance, respectively. The corresponding daily average for each component is subtracted from all the components.

**Figure 11 sensors-23-09428-f011:**
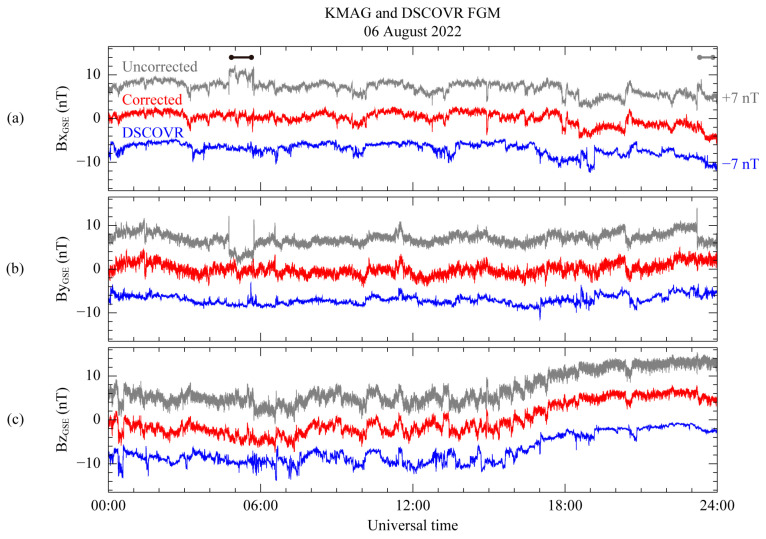
Comparison between the final corrected MAG1 and DSCOVR FGM measurements for 6 August 2022 using GSE coordinate system. (**a**–**c**) Magnetic field components B_X_, B_Y_, and B_Z_, respectively. Corrected MAG1 measurements (red) are shown with initial uncorrected measurements (gray). Blue lines represent measurements from DSCOVR FGM.

**Table 1 sensors-23-09428-t001:** Specifications for the KMAG magnetometer.

Parameter	Performance
Magnetometer type	Fluxgate (racetrack)
Measurable range	±1000 nT
Resolution	<0.2 nT at 10 Hz sampling rate
Mass	3.5 kg
Power	Input: +28 V (unregulated +24–32.8 V)Consumption: 4.6 watt
Operating temperature	KMAG Assy.: −55–70 °CFCE: −20–50 °C
Noise level	<50 pT Hz^−1/2^ at 1 Hz
Axis orthogonality	<1°

## Data Availability

The data that support the findings of this study are available from the corresponding author upon request.

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
