# Peer review of "Correction of Spacecraft Magnetic Field Noise: Initial Korean Pathfinder Lunar Orbiter MAGnetometer Observation in Solar Wind"

_sensors, 2023, doi:10.3390/s23239428_

Round 1
Reviewer 1 Report
Comments and Suggestions for Authors
Please see attached file.

Reviewer 2 Report
Comments and Suggestions for Authors
Dear Authors,
your work is interesting, well structured and the procedure seems giving good results. The cleanliness procedure is not so easy and some improvements on the description might be helpful for the reader to follow the text. In this respect, I suggest the use of some schematic images.
Attached is a commented version of your paper, where I indicated what is not clear and should be corrected/improved.
A point that is not clear is about the high frequency noise: it is described theoretically that amplification of High-Frequency noise can happen when similar data are used for the correction procedure and how to mitigate it, but then it not shown in the experimental case, or at least it is not clear. Please, describe it in detail in paragraph 4.
Last point is about the application of the technique: I understand that it cannot be done once and for all, but instead it has to be applied each time some disturbances from the vessel happens. Please clearly specify it in the introduction and conclusions.

The quality of the English Language is fine in my opinion, I just indicated some terms here and there that might be better clarified.
Round 2
Reviewer 2 Report
Comments and Suggestions for Authors
Dear Authors,
I appreciate that you have taken on board all my comments and suggestions. In my opinion your paper is now clear and straightforward.
I have just one last suggestion about Figure 2. Make it in higher quality or with bigger letters, so as to be able to read also subscrips by zooming it in the pdf version.
Best regards